# Prevalence and intensity of *Schistosoma mansoni* infection, and contributing factors in Alamata district of Tigray Region, Northern Ethiopia

**Gessessew Bugssa**[1,2]*, **Tilahun Teklehaymanot**[2], **Girmay Medhin**[2], **Nega Berhe**[2]

**1** Department of Medical Parasitology and Entomology, College of Health Sciences, Mekelle University, Mekelle, Ethiopia, **2** Aklilu Lemma Institute of Pathobiology, Addis Ababa University, Addis Ababa, Ethiopia

* bugssag@gmail.com

**Data Availability Statement:** All relevant data are within the manuscript and its Supporting Information files.

## Abstract

### Background

Intestinal schistosomiasis caused by *Schistosoma mansoni* continues to be a significant public health problem in Ethiopia. This study investigated the prevalence and intensity of *S. mansoni* infection, and contributing factors in Alamata district of Tigray Region, Northern Ethiopia.

### Methods

A community-based cross-sectional study was conducted and 1762 participants were enrolled from five clusters in Alamata district. A questionnaire was used to collect socio-demographic data and risk factors. Stool samples were examined using Kato-katz technique to determine the prevalence and intensity of infection. The data were analyzed using SPSS version 25. Median, inter quartile range (IQR), mean, frequency, and percentage were used to descriptively summarize data. The Wilcoxon Mann–Whitney and Kruskal-Wallis tests were used to compare the differences in mean rank of egg load between different groups. Bivariate and multivariable logistic regression models were used to investigate the association between the odds of being infected with *S. mansoni* and the different socio-demographic and other factors. The strength of these associations was reported using odds ratio with corresponding 95% confidence intervals, and a P-value below 5% was used to report statistical significance.

### Results

Out of 1762 residents included in the study 941 (53.4%) were females. The age varied from 5–80 years, with a median age of 25 years (IQR = 27), the overall prevalence of *S. mansoni* was 21.5% with males accounting for 26% (204/821) of the infections. The proportion of infection was higher among the age groups of 15–19 and 20–29 years at 32.7% and 33.1%, respectively. The mean egg count among the infected study participants was 146.82 eggs per gram of feces (epg) ± (243.17 SD). Factors significantly associated with increased odds

**Funding:** The author(s) received no specific funding for this work.

**Competing interests:** The authors have declared that no competing interests exist.

of infection were living in Waja cluster (AOR:8.9; 95% CI, 3.5–23.2; P< 0.001); being in the age groups 10–14 (AOR:6.0, 95% CI: 3.1–11.7, P<0.001), 15–19 (AOR:5.8, 95% CI:2.8–12.2, P<0.001), and 20–29 (AOR:3.5, 95% CI:1.8–6.8; P<0.001) years; having direct contact with water while crossing river (AOR: 2.4, 95% CI: 1.5–3.8, P<0.001); and swimming (AOR: 1.4, 95% CI: 1.01–2.0, P = 0.035).

## Conclusion

The study indicates a notable *S.mansoni* burden in the area, driven by various risk factors. To effectively address this, enhancing diagnostics, implementing targeted mass drug administration, and conducting comprehensive health education campaigns on disease transmission routes are imperative.

## Author summary

Schistosomiasis is one of the most prevalent neglected tropical diseases (NTDs) in Ethiopia posing a significant public health burden. Schistosomiasis causes significant morbidity and reduces economic productivity as affected people frequently have chronic health conditions. This study investigated the prevalence of *Schistosoma mansoni* and the risk factors linked with it in the Alamata District of Tigray Region, Northern Ethiopia. This study revealed a considerable prevalence of *S.mansoni* in the local community, notably among children and young adults. The study also identified several contributing factors to the high prevalence of *S.mansoni*. Conditions that promote contact with contaminated water such as proximity to water bodies, having direct contact with water while crossing, swimming habits and poor protective shoe wearing behaviors were among the potential factors associated with risk of *S.mansoni* infection in the Alamata District of Tigray Region. Furthermore, the findings underscore the need for integrated control strategies such as improved water and sanitation, regular health education campaigns, and community-based interventions to reduce the burden of Schistosomiasis in the study area in particular and in the region in general. Overall, this study sheds light on to the epidemiology of *S. mansoni* in Alamata District and highlights the importance of addressing risk factors through targeted public health interventions to improve health outcomes in the affected communities.

## Introduction

Schistosomiasis is an acute and chronic parasitic disease caused by blood flukes of the genus Schistosoma [1,2]. *Schistosoma haematobium*, *Schistosoma mansoni*, and *Schistosoma japonicum* are the three main species that cause human schistosomiasis [2,3]. Humans become infected with schistosomes when they come into contact with fresh water that has been contaminated by cercariae, the infectious stage of the schistosomes that are discharged by the intermediate snail host and that penetrate the intact human skin [2,4–6].

The distribution of schistosomiasis is determined by environmental, biological, demographic, as well as socio-cultural and socio-economic factors that favor the transmission cycle of the parasites [7]. The disease is common in tropical and subtropical regions, particularly in underprivileged areas without access to clean water and proper sanitation facilities [8]. It is

endemic in 78 countries [9] affecting approximately 240 million people worldwide and causing 200,000 deaths annually [1,10,11]. The disease also remains one of the top neglected tropical diseases in Africa causing high morbidity and considerable mortality. This continent accounts for more than 90% of the schistosomiasis infections occurring worldwide [12].

In Ethiopia, both intestinal (caused by *S.mansoni*) and urinary (caused by *S.hematobium*) schistosomiases have been recognized as significant public health issues since the 1960s [13]. Being one of the countries with high burden of schistosomiasis, there are about 5.01 million people who have schistosomiasis and an approximately 37.5 million people are at risk of infection [14] with 3.4 million pre-school children, 12.3 million school-aged children, and 21.6 million adults [15]. Literature documented that intestinal schistosomiasis is widely distributed throughout the country and the prevalence ranges from less than 1% up to more than 90% in some localities [16–27].

Given the high burden of the disease, Ethiopia has been trying to prevent and control the disease since the 1970s [13,28]. However, it is since 2007 that Ethiopia's efforts to prevent and control schistosomiasis and other soil transmitted helminthes (STH) have recorded significant milestones. In 2007, the deworming program was initiated where 1 million school-aged children were treated which was then followed by an expansion in 2013 which reached 1.07 million children. By April 2015, more than 2.9 million children were treated. Moreover, following the launch of the national deworming program by the Federal Ministry of Health in November 2015, nearly 5 million children were treated in that month. Subsequently, a comprehensive five-year national program was established (2016–2020) which aimed to distribute over 100 million treatments and extend coverage to adults in priority areas to effectively control morbidity from schistosomiasis and STH by 2020 [29,30].

Despite ongoing efforts to control schistosomiasis, significant gaps indicate the need for an epidemiological study in Ethiopia in general, and in the current study area in particular. Previous investigations of *S.mansoni* in the study area [18,31] have been limited in scope, focusing on specific population groups, institutional settings, and confined to a known endemic areas with findings that did not accurately represent parasite distribution in the general population and district as a whole. Furthermore, it has been ten years since the last school-based study was conducted in the area. Thus, the gaps in epidemiological data such as outdated prevalence figures, insufficient geographic coverage and representation, and limited analysis of socio-demographic and environmental risk factors in the study area need to be addressed. To address these gaps, we undertook large-scale research to determine the prevalence and risk factors for *S.mansoni* infection in the Alamata area of Tigray Region, Northern Ethiopia. We believe that this study demonstrates a comprehensive understanding of *S.mansoni* epidemiology, which is essential for evidence-based interventions, resource allocation, and policy development to limit the impact of the diseases in the community.

## Methods

### Ethic statement

This study was approved by the Institutional Review Board (IRB) of Aklilu Lemma Institute of Pathobiology of Addis Ababa University under approval number of ALIPB/IRB/006/2017/2018. Further permission was obtained from Tigray Regional Health Bureau, Alamata Woreda Health office, and respective kebele administrations. The intent of the study was elaborated by the data collection teams to each of the study participants after which verbal consent was obtained from all adult participants. For participants under the age of 18, verbal consent was obtained from their guardians, and assent was obtained from the participants themselves. All participants were informed of the study procedures, risks, and benefits, and were aware that

participation was voluntary, and that they could withdraw at any time during the study period. Confidentiality of participant information was maintained throughout the study. Participants who were positive for *S.mansoni* were linked to the nearby health facility for treatment.

## Study setting

Alamata district, located in the southernmost part of the Tigray National Regional State, is situated 600 kilometers north of Addis Ababa and approximately 180 kilometers south of Mekelle, the capital of Tigray. This district borders the Amhara region to the South and West, and the Afar region to the East, Raya Azebo woreda to the Northeast, and Oflla woreda to the North. Alamata encompasses both rural and urban areas, with the rural section known as Raya Alamata, which includes 14 rural kebeles (peasant associations) and one semi urban, and the urban area, Alamata town, comprising 4 kebeles (Fig 1). Kebele (also known as Tabia in Tigrigna) is the smallest administrative unit in Ethiopia. According to the 2007 census, the population of Alamata was estimated to exceed 150,000 [32]. A significant portion of the population is engaged in agriculture, while some urban residents participate in small-scale businesses and services. This region features a warm, semi-arid climate with annual rainfall

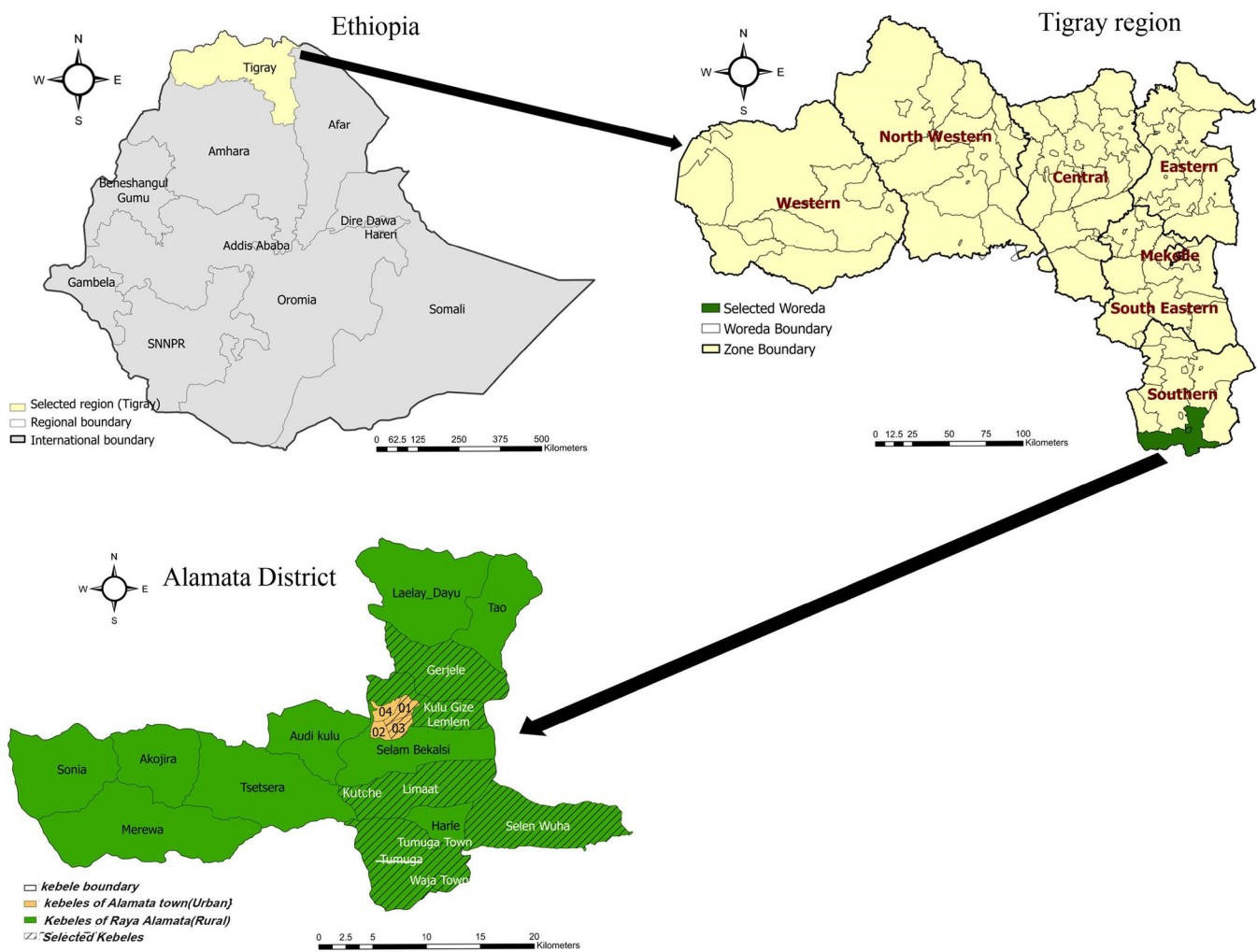

**Fig 1. Map of Alamata district (Map was created using ArcGIS Pro 3; shapefile source is Diva-GIS, available at https://diva-gis.org/).**

ranging from 600 to 800 mm. The altitude varies from 1,178 to 3,148 meters, with 75% classified as lowland below 1,500 meters above sea level (masl). The landscape is characterized by steep, undulating mountains that drain into the Alamata valley, forming a network of gullies that contribute to seasonal rivers. Waja Wuha (also called Gedera River) is the primary stream that flows the whole year round and runs from the southwest (Waja cluster) to the northeast (Selen-Wuha cluster). This river is recognized for its agricultural potential, where farmers cultivate cereals and vegetables and raise livestock. Besides, there are also a number of deep and shallow wells, and ponds used for irrigation developments [33]. According to Health Management Information Systems (HMIS report of the Alamata Woreda Health Office, acute febrile illness, acute upper respiratory infections, pneumonia and parasitic diseases are among the top ten health problems of the district. The area is supported by one hospital, six health centers, and 13 health posts, catering to the health needs of its residents [34,35].

## Study design, objective and study period

A community-based cross-sectional study was conducted to assess the prevalence and intensity of *S.mansoni*, and to investigate factors associated with *S.mansoni* in Alamata District of Tigray Region, Northern Ethiopia. The study was conducted from December 2019 to March 2020.

## Eligibility criteria

All volunteer participants aged 5 years and above and those who were not critically ill at the time of study were included in the study.

## Sample size and Sampling technique

The sample size was estimated using the single population proportion formula as shown below.

$n = \frac{(Z\alpha/2)^2 P(1-P)}{d^2}$, where n is the required sample size, Z is statistics that corresponds to the desired confidence level (95%), P is estimated population proportion, and d is margin of error (the desired precision of the estimate). Taking a prevalence of 73.9% as the best available estimate for the prevalence of *S.mansoni* infection from a previous study [18], a margin of error of 2.1% and a z-score of 1.96, resulted in a sample size of 1681. Adding 5% for non-response, the final sample size was 1765.

The survey utilized a simple random sampling technique to select five clusters from a total of six. The selected clusters included Alamata town, Gerjele, Selen-Wuha, Waja, and Tumuga. Within each cluster, specific kebeles were randomly selected: two from Alamata (Kebele 01 and Kebele 02), two from Gerjele (Kulugize Lemlem and Gerjele); four from Waja (ketena 01, 02, and 05, plus a rural kebele Botamariam), and two from Tumuga (Limat and Kutche). Proportional allocation was used to determine the number of participants from each cluster, with an equal number assigned to each selected kebele in their respective cluster. Each family in the kebele had a folder at their respective health post and/or health center. The sampling fraction (K$^{th}$) value for each kebele was calculated based on the available family folders to systematically select households. Once households were selected, all individuals who met the eligibility criteria were included in the study. This approach ensured a comprehensive and representative sample from each cluster. Finally, the final distribution of study participants was as follows: 174 for Selen-Wuha, 299 for Gerjele, 358 for Tumuga, 804 for Waja, and 130 for Alamata totaling 1,765 participants (Fig 2).

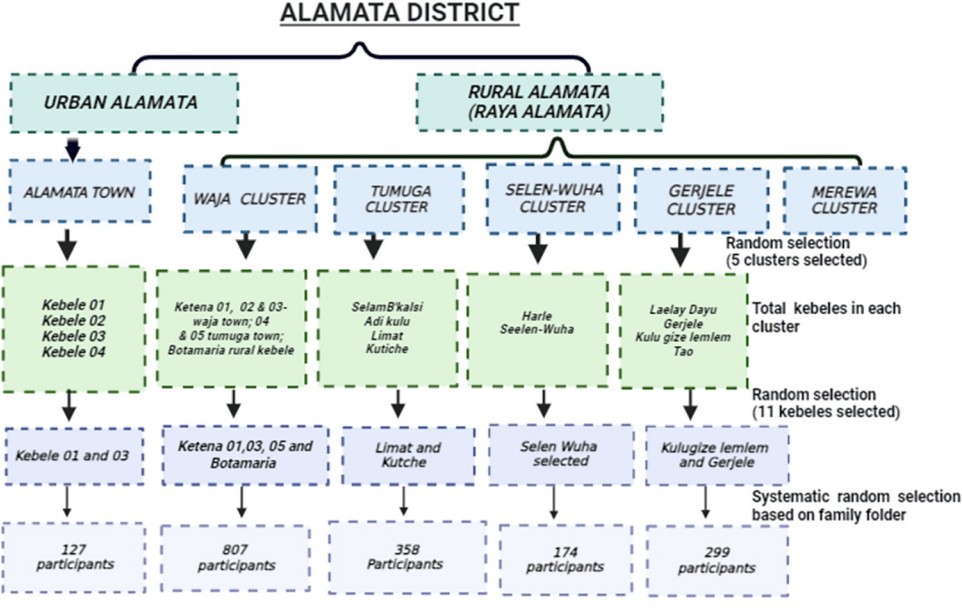

**Fig 2. Schematic presentation of the sampling procedure (Created using BioRender).**

### Data collection procedure

**Questionnaire survey.**   A well-structured questionnaire was developed by adapting from those used in previous studies [17,26,36–38] to obtain socio-demographic information such as age, sex, marital status, educational background etc. and potential risk factors for *S.mansoni* such as history of habit of wearing shoe, bathing and swimming in nearby water bodies, and other activities. The questionnaire was originally written in English, and then translated into Tigrigna (local language). Data was collected by trained BSc nurses and Medical Laboratory technologists. All data collectors were instructed on the importance of confidentiality and ethical considerations during the data collection process.

**Parasitological investigation.**   We approached the selected households via the Health Extension Workers (trained community-based health professionals who deliver primary healthcare services and are assigned to each kebele, familiar with the details of the community). Following an interview, every eligible individual in each selected household was then provided with a dry, clean, labeled plastic cup and spatula to collect approximately 3g (roughly equivalent to the size of a matchstick head) of fecal samples. Collected stool samples were then transported to our nearby temporarily established centers (health post or health center) for processing and preparation of the Kato-Katz smears. By employing the Kato-Katz thick smear approach [39], duplicate Kato thick smears were prepared for every subject utilizing a template that delivered 41.7 mg of feces. The Kato-Katz slides were then properly packed and transported to the Medical Microbiology and Parasitology Laboratory of the College of Health Sciences, Mekelle University for examination. The Kato-Katz slides were examined with a light microscope using the 10x and 40x objective lenses by Medical Parasitologists to investigate and quantify the ova of *S.mansoni*. The average egg count of the two slides was then multiplied by a factor of 24 to obtain the number of eggs per gram (epg) of stool. The *S.mansoni* intensity was classified based on WHO criteria, where 1−99 eggs per gram (EPG) is light infection, 100−399 EPG is moderate infection, and ≥400 is heavy infection [40].

## Data Quality Assurance and management

To ensure data quality, the questionnaire was pretested, and the information obtained was used to improve it. The data collectors underwent a three-day training session covering the study protocol, obtaining informed consent, appropriate participant interaction, ethical procedures, and specimen collection. The data collection teams were closely followed, gaps were addressed, and open communication was maintained with the data collectors to ensure that the data collection remained on track. Nurse professionals conducted the interview; Medical Laboratory Technologists were responsible for sample collection and Kato-Katz smear preparation; while Medical Parasitologists conducted the microscopic investigation of the parasites.

## Data analysis

After the data was collected, it was cleaned, coded, and entered into EpiData version 3.0, and exported to SPSS statistical software for Windows, version 25 (SPSS, Chicago, IL, USA) for analysis. Inconsistent and incomplete values were checked against the filled questionnaire and corrected as necessary. Data consistency and completeness were checked using exploratory data analysis. First, univariate analyses of each of the measures were computed and checked for data irregularities. Median, Inter quartile range (IQR), mean, frequency, and graphs were used to descriptively summarize the data. Data were checked for normality and log-transformed [log10] to reduce the skewness of a measurement variable. In this manuscript, only non-transformed means were reported. Because of the over-dispersion of egg counts (were not normally distributed), the Wilcoxon Mann–Whitney and Kruskal-Wallis tests were used to compare the differences in the median egg load (intensity of infection) between the different socio-demographic characteristics. For bivariate analysis, statistical significance of categorical variables was measured by the chi-square test or Fischer's exact test as proper. Variables with a chi-squared p-value of less than 25% were included in the multivariable models. Results were reported as statistically significant when the p-value was less than 5%. Eventually, multivariable analyses were done to identify the independent effect of the main explanatory variable on *S. mansoni* infection outcomes of interest after adjusting for confounding variables. Odds ratio was computed with 95% confidence intervals as a measure of the strength of the association.

## Operational definitions

Kebele: the smallest administrative unit in the Ethiopian administrative hierarchy and can be used in rural as well as urban areas.

Ketena: is less known and slightly smaller than kebele and is used most of the time in the urban areas.

Cluster: the term "cluster" denotes a local administrative framework utilized in the health sector of the study area, which combines multiple kebeles into a larger unit created for ease of administration. Typically, each cluster consists of one main health center and one or more health posts that serve the population within its designated area (depending on the size of the cluster). A cluster usually encompasses two to four adjacent kebeles, depending on the local administrative structure. In essence, a cluster is larger than a kebele.

## Results

### Socio-demographic characteristics

A total of 1672 participants were included in this study with a response rate of 99.8%. The median age of study participants was 25 years (IQR = 27) and 53.4% were females. Nearly 90% were followers of Orthodox Christianity, 45.3% were single and/or underage, 39.5% were

**Table 1. Socio-demographic characteristics of participants in Alamata district of Tigray region, Northern Ethiopia (n = 1762).**

| Socio-demographic characteristics | | Frequency | Percent [95% CI] |
|---|---|---|---|
| **Age category** | 5–9 | 189 | 10.7 [9.3, 12.2] |
| | 10–14 | 397 | 22.5 [20.6, 24.5] |
| | 15–19 | 145 | 8.2 [7.0, 9.6] |
| | 20–29 | 282 | 16.0 [14.3, 17.8] |
| | 30–39 | 264 | 15.0 [13.4, 16.7] |
| | 40–49 | 227 | 12.9 [11.4, 14.5] |
| | 50–59 | 115 | 6.5 [5.4, 7.8] |
| | > = 60 | 143 | 8.1 [6.9, 9.5] |
| **Sex** | Male | 821 | 46.6 [44.3, 48.9] |
| | Female | 941 | 53.4 [51.1, 55.7] |
| **Religion** | Orthodox | 1565 | 88.8 [87.3, 90.2] |
| | Muslim | 190 | 10.8 [9.4, 12.3] |
| | Others | 7 | 0.4 [0.2, 0.8] |
| **Marital status** | Single and/or underage | 798 | 45.3 [43.0, 47.6] |
| | Married/Cohabiting | 696 | 39.5 [37.2, 41.8] |
| | Divorced | 160 | 9.1 [7.8, 10.5] |
| | Widowed | 108 | 6.1 [5.1, 7.3] |
| **Occupation** | Employed | 92 | 5.2 [4.3, 6.3] |
| | Merchant | 90 | 5.1 [4.2, 6.2] |
| | Farmer | 489 | 27.8 [25.7, 29.9] |
| | Housewife | 225 | 12.8 [11.3, 14.4] |
| | Daily laborer | 55 | 3.1 [2.4, 4.0] |
| | Student | 720 | 40.9 [38.6, 43.2] |
| | Unemployed | 91 | 5.2 [4.2, 6.3] |
| **Educational Status** | Illiterate | 588 | 33.4 [31.2, 35.6] |
| | Elementary (1–8 grade) | 943 | 53.5 [51.2, 55.8] |
| | Secondary & high school(9–12 grade) | 147 | 8.3 [7.1, 9.7] |
| | College and above | 84 | 4.8 [3.8, 5.8] |
| **Cluster** | Alamata | 127 | 7.2 [6.1, 8.5] |
| | Gerjele | 299 | 17.0[15.3, 18.8] |
| | Selen-Wuha | 174 | 9.9 [8.5, 11.3] |
| | Tumuga | 358 | 20.3 [18.5, 22.2] |
| | Waja | 804 | 45.6 [43.3, 48.0] |
| **Residence** | Urban | 636 | 36.1 [33.9, 38.4] |
| | Rural | 1126 | 63.9 [61.6, 66.1] |

currently married, most common occupational status was being a student (40.9%), more than half (53.5%) attended elementary school (1-8[th] grade), and about two-thirds (63.9%) were from rural areas (Table 1).

## Distribution of *S.mansoni* infection with respect to different socio-demographic factors

The overall prevalence of *S.mansoni* was 21.5% with nearly one-quarter of the male population in the study were affected by *S.mansoni*. Besides, the proportion of infection was higher among participants in the age groups 10–14 and 20–29 years which accounted for 32.7% and 33.1%, respectively (Table 2).

**Table 2. Distribution of *S.mansoni* and intensity of infection with respect to Socio-demographic characteristics of study participants in Alamata district of Tigray region, Northern Ethiopia, (N = 1762).**

| Socio-demographic variables | | Positive for *S.mansoni* | | Intensity of infection | | |
|---|---|---|---|---|---|---|
| | | yes | no | Light | Moderate | Heavy |
| Age | 5–9 | 33(17.5%) | 156(82.5%) | 19(57.6%) | 10(30.30%) | 4(12.10%) |
| | 10–14 | 130(32.7%) | 267(67.3%) | 86(66.2%) | 34(26.2%) | 10(7.7%) |
| | 15–19 | 48(33.1%) | 97(66.9%) | 29(60.4%) | 15(31.3%) | 4(8.3%) |
| | 20–29 | 58(20.6%) | 224(79.4%) | 39(67.2%) | 18(31.0%) | 1(8.3%) |
| | 30–39 | 38(14.4%) | 226(85.6%) | 24(63.2%) | 12(31.6%) | 2(5.3%) |
| | 40–49 | 39(17.2%) | 188(82.8%) | 33(84.6%) | 5(12.8%) | 1(2.6%) |
| | 50–59 | 14(12.2%) | 101(87.8%) | 9(64.3%) | 5(35.7%) | 0(0.0%) |
| | > = 60 | 19(13.3%) | 124(86.7%) | 10(52.6%) | 7(36.8%) | 2(10.5%) |
| Sex | Male | 204(24.8%) | 617(75.2%) | 129 (63.2%) | 55(27.0%) | 20(9.8%) |
| | Female | 175 (18.6%) | 766 (81.4%) | 120(68.6%) | 51(29.1%) | 4(2.3%) |
| Educational status | illiterate | 94(16.0%) | 494(84.0%) | 60(63.8%) | 29(30.9%) | 5(5.3%) |
| | Elementary(1–8 grade) | 235(24.9%) | 708(75.1%) | 152(64.7%) | 65(27.7%) | 18(7.7%) |
| | Secondary & high school | 38(25.9%) | 109(74.1%) | 26(68.4%) | 11(28.9%) | 1(2.6%) |
| | College and above | 12(14.3%) | 72(85.7%) | 11(91.7%) | 1(8.3%) | 0(0.0%) |
| Religion | Orthodox | 340(21.7%) | 1225(78.3%) | 221(65.0%) | 99(29.1%) | 20(5.9%) |
| | Muslim | 37(19.5%) | 153(80.5%) | 26(70.3%) | 7(18.9%) | 4(10.8%) |
| | Others | 2(28.6%) | 5(71.4%) | 2(100.0%) | 0(0.0%) | 0(0.0%) |
| Occupation | Employed | 18(19.6%) | 74(80.4%) | 14 (77.8%) | 3(16.7%) | 1(5.6%) |
| | Merchant | 16(17.8%) | 74(82.2%) | 11(68.8%) | 5(31.3%) | 0(0.0%) |
| | Farmer | 76(15.5%) | 413(84.5%) | 50(65.8%) | 22(28.9%) | 4(5.3%) |
| | Housewife | 30(13.3%) | 195(86.7%) | 20(66.7%) | 10(33.3%) | 0(0.0%) |
| | Daily laborer | 7(12.7%) | 48(87.3%) | 6(85.7%) | 1(14.3%) | 0(0.0%) |
| | Student | 206(28.6%) | 514(71.4%) | 130(63.1%) | 58(28.2%) | 18(8.7%) |
| | Job seeker | 26(28.6%) | 65(71.4%) | 18(69.2%) | 7(26.9%) | 1(3.8%) |
| Marital status | Single | 228(28.6%) | 570(71.4%) | 143(62.7%) | 66(28.9%) | 19(8.3%) |
| | Married | 106(15.2%) | 590(84.8%) | 74(69.8%) | 29(27.4%) | 3(2.8%) |
| | Divorced | 30(18.8%) | 130(81.3%) | 23(76.7%) | 6(20.0%) | 1(3.3%) |
| | Widowed | 15(0.9%) | 93(5.3%) | 9(60.0%) | 5(33.3%) | 1(6.7%) |

The prevalence of *S.mansoni* was stratified by different clusters of the study area and greater proportion 40.7% of the cases were attributed to the Waja cluster which mainly included two semi-urban areas and one rural kebele followed by the Selen Wuha rural cluster which constituted 13.8% of the infections (Fig 3). The Clusters that were far away from a stream called Waja Wuha (the main stream in the district) had low prevalence of *S.mansoni* than those who were close to the stream such as Waja and Selen Wuha which carried a higher proportion of the infection.

The arithmetic mean egg count among the *S.mansoni* positives was found to be 146.82 epg ± (243.17 standard deviations). Of those who were positive for *S.mansoni*, the majority (65.7%) had a light infection followed by 106 (28.0%) and 24(6.3%) who had a moderate and a heavy infection, respectively; and the maximum egg output per gram of stool (epg) was recorded at 2400. The 5–9 years age group showed a relatively high intensity of moderate to heavy infections, with the egg load peaked in this specific age group. Furthermore, the mean egg load among the infected participants revealed a gradually decreasing trend with age group from 5–9 years with 226.9 epg to the 40–49 years age group to approximately 81epg. However, a sharp increment was noted in the older age groups which reached as high as 174 epg.

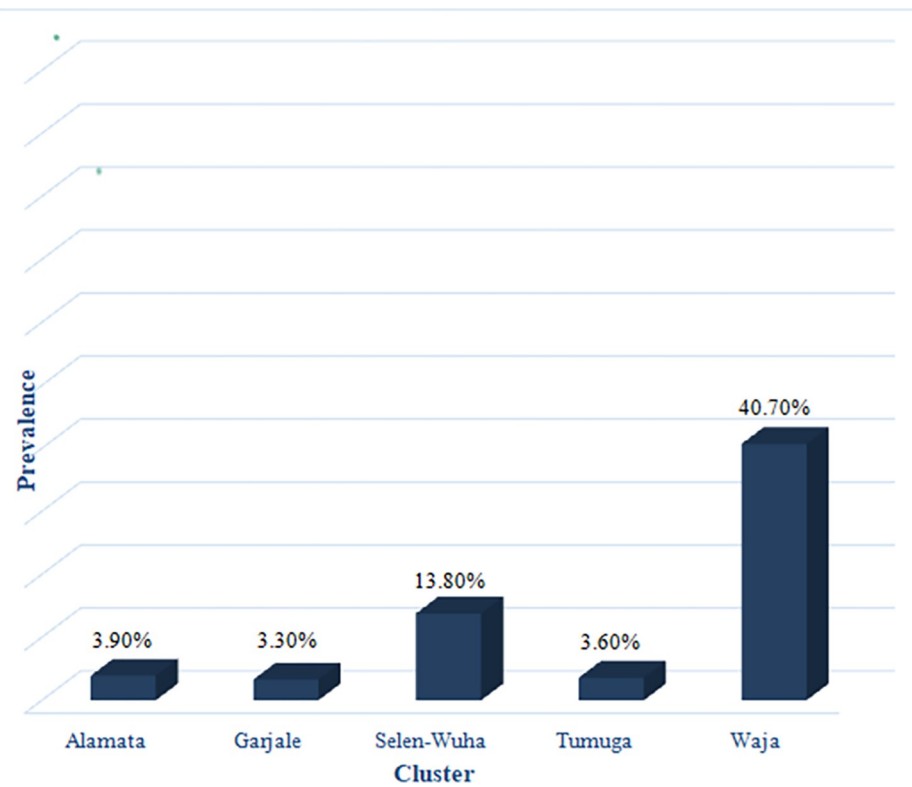

**Fig 3. Distribution of *S.mansoni* by cluster.**

## Distribution of *S.mansoni* infection with respect to water-related factors

Nearly half (853) of the study participants reported bathing in a river or stream, while about 22.0% indicated that they had swum in nearby water bodies. Additionally, half of the participants stated that they washed clothes and/or utensils in a river, and around 10% were involved in irrigation activities. Among those who reported bathing, 31% were infected with *S.mansoni*, and 43.7% of those who swam in nearby rivers or water bodies were also infected. Furthermore, it was found that 29.6% of participants who regularly washed their clothes and utensils in a river were infected with *S.mansoni* (Table 3).

Findings from the Mann-Whitney U test showed no statistically significant difference in egg load (infection intensity) between males and females (Z-value = -1.23, P = 0.254). On the other hand, the findings from Kruskal-Wallis test showed that there was significant difference in median intensity of infection ($\chi 2$ = 9.742, P = 0.045) in the Waja cluster (Table 4).

## Factors associated with *S.mansoni* infection

Variables that include district, age category, gender, educational status, swimming habit, river contact while crossing a river, washing clothes/utensils, and habit of bathing in a river/nearby water body were considered for multiple logistic regression to identify independent predictors of *S.mansoni* infection. Nevertheless, only four of the variables: district, age, crossing rivers/direct contact, and swimming were shown to be significantly associated with *S.mansoni* infection in the multivariable logistic regression analysis after controlling for potential confounders. Accordingly, the odds of being infected with *S.mansoni* among individuals in Waja cluster was nearly nine times (Adjusted Odds Ratio (AOR) = 8.9; 95% CI 3.5–23.2, P< 0.001) as compared

**Table 3. Distribution of *S.mansoni* infection in relation to water contact pattern of participants in Alamata district of Tigray region, Northern Ethiopia (n = 1762).**

| Water-related activity | | Presence of *S.mansoni* | | Total |
|---|---|---|---|---|
| | | Positive | Negative | |
| Bathing body in a nearby river | yes | 264 (31.0%) | 589(69.0%) | 853 |
| | no | 115(12.7%) | 794(87.3%) | 909 |
| Swimming in nearby waterbodies | yes | 170(43.7%) | 219(56.3% | 389 |
| | no | 209(15.2%) | 1164(84.8%) | 1373 |
| wash clothes or utensils in a river | yes | 262(29.6%) | 624(70.4%) | 886 |
| | no | 117(13.4%) | 759(86.6%) | 876 |
| involved in irrigation activities | yes | 59(33.1%) | 119(66.9%) | 178 |
| | No | 320(20.2%) | 1264(79.8%) | 1584 |
| Direct contact with water while irrigating | yes | 57(33.9%) | 111(66.1%) | 168 |
| | no | 322(20.2%) | 1272(79.8%) | 1594 |
| Cross river/direct contact with water while crossing | yes | 352(26.6%) | 972(73.4%) | 1324 |
| | no | 27(6.2%) | 411(93.8%) | 438 |

to individuals in Alamata cluster. Besides, the odds of *S.mansoni* infection among participants who belonged to the age group of 10–14 years was 6 times higher (AOR: 6.0, 95% CI: 3.1–11.7, P<0.001) as compared to the to age group 60 years and above. Besides, the odds acquiring *S. mansoni* infection among individuals who used to swim and who crossed river barefoot was

**Table 4. Comparison of egg load of *S.mansoni* among different groups of Socio-demographic variables using Kruskal-Wallis test in Alamata district of Tigray region, Northern Ethiopia (N = 379).**

| Group | | Frequency | Mean rank | Chi-Square | Df | P value |
|---|---|---|---|---|---|---|
| Educational status | illiterate | 94 | 182.56 | 3.840 | 3 | 0.279 |
| | Elementary (1–8 grade) | 235 | 196.08 | | | |
| | secondary & high school | 38 | 186.64 | | | |
| | College and above | 12 | 139.83 | | | |
| Age category | 5–9 | 33 | 214.20 | 12.356 | 7 | 0.089 |
| | 10–14 | 130 | 196.14 | | | |
| | 15–19 | 48 | 207.47 | | | |
| | 20–29 | 58 | 178.86 | | | |
| | 30–39 | 38 | 186.41 | | | |
| | 40–49 | 39 | 142.97 | | | |
| | 50–59 | 14 | 175.75 | | | |
| | > = 60 | 19 | 210.05 | | | |
| Occupation | Student | 206 | 187.32 | 5.711 | 5 | 0.335 |
| | Employed | 18 | 163.97 | | | |
| | Merchant | 16 | 155.03 | | | |
| | Farmer | 76 | 167.05 | | | |
| | Housewife | 30 | 157.88 | | | |
| | Daily laborer | 7 | 147.00 | | | |
| Cluster | Alamata | 5 | 162.10 | 9.742 | 4 | 0.045* |
| | Gerjele | 10 | 95.40 | | | |
| | Selen Wuha | 24 | 171.79 | | | |
| | Tumuga | 13 | 175.62 | | | |
| | Waja | 327 | 195.23 | | | |

**Table 5. Multivariable factor analysis associated with *S.mansoni* infection in Alamata district of Tigray region, Northern Ethiopia.**

| Predictors | | Schistosoma infection, n (%) | | Bivariate | | Multivariate | |
|---|---|---|---|---|---|---|---|
| | | No (%) | Yes (%) | COR [95% CI] | P-value | AOR [95% CI] | P-value |
| Clusters | Alamata | 122 (96.1) | 5 (3.9) | 1 | | 1 | |
| | Gerjele | 289 (96.7) | 10 (3.3) | 0.8 [0.3, 2.5] | 0.762 | 0.6[0.2, 1.9] | 0.377 |
| | Selen wuha | 150 (86.2) | 24 (13.8) | 3.9 [1.4, 10.5] | 0.007 | 2.1 [0.8, 6.0] | 0.154 |
| | Tumuga | 345 (96.4) | 13 (3.6) | 0.9 [0.3, 2.6] | 0.876 | 0.4 [0.1, 1.3] | 0.135 |
| | Waja | 477 (59.3) | 327 (40.7) | 16.7 [6.8,41.1] | <0.001 | 8.9 [3.5, 23.2] | <0.001 |
| Age category (years) | 5–9 | 156 (82.5) | 33 (17.5) | 1.4 [0.7, 2.5] | 0.301 | 2.1[1.0, 4.2] | 0.046 |
| | 10–14 | 267 (67.3) | 130 (32.7) | 3.2 [1.9, 5.4] | <0.001 | 6.0 [3.1, 11.7] | <0.001 |
| | 15–19 | 97 (66.9) | 48 (33.1) | 3.2 [1.8, 5.8] | <0.001 | 5.8[2.8, 12.2] | <0.001 |
| | 20–29 | 224 (79.4) | 58 (20.6) | 1.7 [1.0, 3.0] | 0.068 | 3.5 [1.8, 6.8] | <0.001 |
| | 30–39 | 226 (85.6) | 38 (14.4) | 1.1 [0.6, 2.0] | 0.759 | 1.7 [0.9, 3.3] | 0.100 |
| | 40–49 | 188 (82.8) | 39 (17.2) | 1.4 [0.7, 2.5] | 0.317 | 1.8 [1.0, 3.5] | 0.062 |
| | 50–59 | 101 (87.8) | 14 (12.2) | 0.9 [0.4, 1.9] | 0.790 | 1.3 [0.6, 2.9] | 0.484 |
| | 60 or above | 124 (86.7) | 19 (13.3) | 1 | | 1 | |
| Sex | Male | 617 (75.2) | 204 (24.8) | 1.4 [1.2, 1.8] | 0.001 | 1.2 [0.8, 1.5] | 0.350 |
| | Female | 766 (81.4) | 175 (18.6) | 1 | | 1 | |
| Educational background | Illiterate | 494(84.01) | 94(15.99) | 1.4[0.6, 2.2] | 0.689 | 1.7[0.8, 3.6] | 0.171 |
| | Elementary | 708(75.08) | 235(24.92%) | 2[1.1, 3.7] | 0.032 | 1.6[0.8, 3.4] | 0.227 |
| | Secondary&High School | 109(74.15) | 38(25.85) | 2.1[1.0, 4.3] | 0.043 | 1.6[0.7,3.6] | 0.281 |
| | College & above | 72(85.7%) | 12(14.3%) | 1 | | 1 | |
| Bath body in a river | Yes | 589 (69.1) | 264 (30.9) | 3.1 [2.4, 3.9] | <0.001 | 1.6 [1.1, 2.5] | 0.069 |
| | No | 794 (87.3) | 115 (12.7) | 1 | | 1 | |
| Wash utensils/clothes in a river | Yes | 624 (70.4) | 262 (29.6) | 2.7 [2.1, 3.5] | <0.001 | 0.7 [0.4, 1.0] | 0.068 |
| | No | 759 (86.6) | 117 (13.4) | 1 | | 1 | |
| Swim in a water body | Yes | 219 (56.3) | 170 (43.7) | 4.3 [3.4, 5.5] | <0.001 | 1.4 [1.1, 2.0] | 0.035 |
| | No | 1164 (84.8) | 209 (15.2) | 1 | | 1 | |
| Involved in an irrigation activity | Yes | 119 (66.9) | 59 (33.1) | 2.0 [1.4, 2.7] | <0.001 | 1.4 [0.9, 2.1] | 0.159 |
| | No | 1264 (79.8) | 320 (20.2) | 1 | | 1 | |
| Crossing river | Bare foot/direct contact with river | 981 (73.6) | 352 (26.4) | 5.3 [3.6, 8.0] | <0.001 | 2.4 [1.5, 3.8] | <0.001 |
| | Use protective shoe/bridge | 402 (93.7) | 27(6.3) | 1 | | 1 | |

2.4 (AOR; 2.4, 95% CI: 1.5, 3.8, P<0.001) and 1.4 (AOR: 1.4, 95% CI: 1.1, 2.0, P = 0.035 times as compared to their to their counter parts, respectively (Table 5).

## Discussion

Ethiopia, in partnership with several international organizations, has made significant efforts to combat schistosomiasis through integrated approaches that combine mass drug administration (MDA) with water, sanitation, and hygiene (WASH) as well as behavioral change communication to promote healthy practices [28,29,30,41,42]. As part of these efforts, the country launched a school-based deworming campaign in 2015, aiming to reach 17 million children [43]. Despite these efforts, the prevalence of *S.mansoni* infection remains a public health problem in Ethiopia [14,15]. In line with this, the overall prevalence of *S.mansoni* in this study was found to be 21.5% indicating that the parasite continued to be a public health problem in the study area. According to the WHO endemicity (threshold) classification of *S.mansoni* infection in a community, it is considered moderate lying between 10% and < 50% [10]. The

finding of this study goes in agreement with other studies conducted in the Northwest [21] and South [27] of Ethiopia where prevalence rates were reported between 20.0% and 22.0%.

In contrast, the results of the current study showed a lower prevalence of *S.mansoni* infection compared to the rates reported in previous studies conducted in various parts of Ethiopia such as 52.1% in Kemsie [44] 63.0% in Adwa [17] and 89.9% in Sanja [45]. To this end, the findings of this study even varied from a study conducted 10 years earlier in the same study area where the prevalence was 73.9% [18]. Although the literature has documented that variations can arise even within a single village in *S.mansoni* endemic areas [46], the primary cause for such discrepancies may be attributed to the fact that this study covered a wider geographic area and older age groups (>15 years) than the previous study which might have diluted our results. Besides, sample size, time of study, increased participants' awareness through time, and current control measures could have potentially contributed to the observed discrepancies. The findings of this study were also lower than studies conducted elsewhere in Africa such as in The Democratic Republic of the Congo, Uganda, Cote d'Ivoire, and Tanzania with prevalence ranging from 36.0% to 84% [46–50].

On the contrary, lower magnitudes of *S.mansoni* ranging from 0.6% to approximately 17% were recorded in different corners of Ethiopia such as Hintalo-Wejerat district in the North [51]; Amibera district in the Northeast [52]; Libo-kemkem in the Northwest [53]; Dawro in the South [38]; Gilgel-Gibe district in the South West [54]; Guder [55] in the West; and Shoa-Robbit in the North Central parts of Ethiopia [25]. This was found to be higher than studies conducted elsewhere such as Sudan, Kenya, Tanzania, Nigeria, and Ghana with varying prevalence reported between 1.9% and 19.8% [56–61]. Such discrepancies could be attributed to factors such as variations in the abundance of water bodies and distribution of snail intermediate hosts, environmental conditions, socio-economic disparities, differences in the study periods and diagnostic methods, as well as variations in the composition of the study populations [14]. This study also attempted to assess the association of different socio-demographic and water contact activities of the study participants with the prevalence and intensity of infection of *S.mansoni*. Consequently, it was depicted that *S.mansoni* infection was found to be significantly associated with the age groups 10–14 (AOR: 6.0, 95% CI: 3.1–11.7, P<0.001), 15–19 (AOR: 5.8, 95% CI: 2.8–12.2, P<0.001), and 20–29 years (AOR: 3.5, 95% CI: 1.8–6.8; P<0.001). This might be explained by the fact that participants in these age groups have frequent water contact as part of recreational (swimming and bathing) or occupational (irrigation) activities which increase their exposure to infected water sources. Similar trends have been reported in studies from North and Northwest Ethiopia, as well as other parts of Africa [17,62,63]. Conversely, the egg load decreased as the age group increased, which may be explained by lower exposure to contaminated water, development of acquired immunity, and resistance to re-infection among adults [17,50].

Among the potential contributing factors investigated, the prevalence of *S.mansoni* infection was also found to be significantly associated with water contact behaviors such as swimming habits and river contact with barefoot while crossing. In line with this, the prevalence of intestinal schistosomiasis and water contact behavior has been extensively investigated and the association is well established by various authors [20,22,26,64].

The prevalence and intensity of infection varied by cluster with the highest prevalence and intensity of infection recorded in Waja. A plausible explanation for this marked variation would be that the proximity of this cluster to a river known as the Waja wuha (also locally called Gedera) and residents of this cluster had a higher probability for repeated exposure to contaminated water which could increase their chance of contracting infection. This is in line with a study conducted in Uganda, which suggested that individuals residing near water bodies had a higher likelihood of being infected with *S.mansoni* [65]. This is also substantiated by

evidence conducted elsewhere [49] that proximity to water bodies was noted to influence the transmission of schistosomiasis.

## Limitations

The limitation of this study is the use of a single stool sample (though duplicate slides were used) which might have resulted in an underestimation of the prevalence and severity of *S. mansoni* infection as daily parasite egg excretion varies. The additional or repetition of sample takings from the subjects was not possible because of the resource limitations.

## Conclusion

The current study reveals a significant burden of *S.mansoni* infection in the study area with prevalence of 21.5%. Key demographic factors, particularly children (5–19 years), and those younger adults (20–29 years) which are part of the productive age group were significantly infected with *S.mansoni*. Besides, behavioral activities such as swimming and crossing rivers barefoot resulting in direct contact with contaminated water were risk factors for *S.mansoni* infection. Furthermore, clusters which were close to a river exhibited a disproportionately high infestation rates. To address this public health challenge, a multifaceted approach is recommended, including enhancing diagnostics, strengthening surveillance, and implementing targeted mass drug administration. Improving access to safe water and sanitation, as well as conducting comprehensive health education campaigns, are also crucial to effectively prevent and control schistosomiasis in the community.

## Supporting information

**S1 File. Questionnaire to assess socio-demographic, behavioral or environmental factors associated with *Schistoma mansoni* and infection.**
(DOCX)

**S2 File. Laboratory result reporting format for parasitological findings.**
(DOCX)

**S1 Data. Excel data showing main findings.**
(XLSX)

## Acknowledgments

We would like to express our gratitude to Aklilu Lemma Institute of Pathobiology, Addis Ababa University for giving us the opportunity to undertake this research. We also extend our appreciation to Mekelle University, College of Health Sciences, for giving us the access to its laboratory facilities for the analysis.

## Author Contributions

**Conceptualization:** Gessessew Bugssa, Tilahun Teklehaymanot, Nega Berhe.

**Data curation:** Gessessew Bugssa, Girmay Medhin, Nega Berhe.

**Formal analysis:** Gessessew Bugssa, Girmay Medhin, Nega Berhe.

**Investigation:** Gessessew Bugssa, Nega Berhe.

**Methodology:** Gessessew Bugssa, Tilahun Teklehaymanot, Girmay Medhin, Nega Berhe.

**Supervision:** Tilahun Teklehaymanot, Nega Berhe.

**Visualization:** Gessessew Bugssa, Tilahun Teklehaymanot, Girmay Medhin, Nega Berhe.

**Writing – original draft:** Gessessew Bugssa.

**Writing – review & editing:** Gessessew Bugssa, Tilahun Teklehaymanot, Girmay Medhin, Nega Berhe.

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
