## [Decision Letter · Decision Letter 0]

1 Aug 2024

Dear Mr Bugssa,

Thank you very much for submitting your manuscript "Prevalence and intensity of Schistosoma mansoni infection, and contributing factors in Alamata district of Tigray Region, Northern Ethiopia." for consideration at PLOS Neglected Tropical Diseases. As with all papers reviewed by the journal, your manuscript was reviewed by members of the editorial board and by several independent reviewers. In light of the reviews (below this email), we would like to invite the resubmission of a significantly-revised version that takes into account the reviewers' comments. 

We cannot make any decision about publication until we have seen the revised manuscript and your response to the reviewers' comments. Your revised manuscript is also likely to be sent to reviewers for further evaluation.

Sincerely,

Uwem Friday Ekpo, PhD

Academic Editor

Jong-Yil Chai

Section Editor

Reviewer's Responses to Questions

**Key Review Criteria Required for Acceptance?**

**Methods**

-Are the objectives of the study clearly articulated with a clear testable hypothesis stated?

-Is the study design appropriate to address the stated objectives?

-Is the population clearly described and appropriate for the hypothesis being tested?

-Is the sample size sufficient to ensure adequate power to address the hypothesis being tested?

-Were correct statistical analysis used to support conclusions?

-Are there concerns about ethical or regulatory requirements being met?

Reviewer #1: Yes

Reviewer #2: (No Response)

**Results**

-Does the analysis presented match the analysis plan?

-Are the results clearly and completely presented?

-Are the figures (Tables, Images) of sufficient quality for clarity?

Reviewer #1: Yes

Reviewer #2: (No Response)

**Conclusions**

-Are the conclusions supported by the data presented?

-Are the limitations of analysis clearly described?

-Do the authors discuss how these data can be helpful to advance our understanding of the topic under study?

-Is public health relevance addressed?

Reviewer #1: Yes

Reviewer #2: (No Response)

**Editorial and Data Presentation Modifications?**

Reviewer #1: (No Response)

Reviewer #2: (No Response)

**Summary and General Comments**

Reviewer #1: Introduction:

A brief explanation of current control efforts for schistosomiasis in Ethiopia would strengthen the introduction. Mentioning specific interventions would help frame the study within the national strategy.

Methodology:

1. Study setting:

o "Cluster" likely refers to administrative units like sub-districts or villages. The authors should clarify the specific administrative structure used in the study.

2. Sample size and sampling strategy:

o The authors should double-check their sample size calculation.

o Random selection of five clusters: This likely considered subdistricts as clusters and uses a random sampling method from a sampling frame (a list of all clusters in the study area). The authors need to specify this frame and state the sampling strategy.

o Random selection of 12 kebeles: "Kebeles" may be the lowest administrative units in Ethiopia, like villages, is it correct? Again, a random sampling method was likely used, but clarification is needed.

o If there's a third sampling step, the authors should definitely explain it for transparency.

Discussion:

• Interpreting the findings in light of the national control strategy would be valuable. Here are some specific points to consider:

o Age groups targeted: It's unclear if the national Mass Drug Administration (MDA) campaigns target all ages or just school children. The study can highlight this gap if the Ministry of Public Health doesn't cover all age groups.

o Focal prevalence: The varying prevalence across sub-districts is a crucial finding. The study can emphasize how it suggests "focal prevalence" (infection concentrated in specific areas) and recommend targeting control activities accordingly. This could be at sub-district level depending on the severity of focalization.

Reviewer #2: (No Response)

PLOS authors have the option to publish the peer review history of their article (what does this mean?). If published, this will include your full peer review and any attached files.

Reviewer #1: No

Reviewer #2: No
---

## [Decision Letter · Decision Letter 1]

12 Nov 2024

Dear Mr Bugssa,

We are pleased to inform you that your manuscript 'Prevalence and intensity of Schistosoma mansoni infection, and contributing factors in Alamata district of Tigray Region, Northern Ethiopia.' has been provisionally accepted for publication in PLOS Neglected Tropical Diseases.

Best regards,

Uwem Friday Ekpo, PhD

Academic Editor

Jong-Yil Chai

Section Editor

Shaden Kamhawi

co-Editor-in-Chief

Paul Brindley

co-Editor-in-Chief

Reviewer's Responses to Questions

**Key Review Criteria Required for Acceptance?**

**Methods**

-Are the objectives of the study clearly articulated with a clear testable hypothesis stated?

-Is the study design appropriate to address the stated objectives?

-Is the population clearly described and appropriate for the hypothesis being tested?

-Is the sample size sufficient to ensure adequate power to address the hypothesis being tested?

-Were correct statistical analysis used to support conclusions?

-Are there concerns about ethical or regulatory requirements being met?

Reviewer #1: (No Response)

**Results**

-Does the analysis presented match the analysis plan?

-Are the results clearly and completely presented?

-Are the figures (Tables, Images) of sufficient quality for clarity?

Reviewer #1: (No Response)

**Conclusions**

-Are the conclusions supported by the data presented?

-Are the limitations of analysis clearly described?

-Do the authors discuss how these data can be helpful to advance our understanding of the topic under study?

-Is public health relevance addressed?

Reviewer #1: (No Response)

**Editorial and Data Presentation Modifications?**

Reviewer #1: (No Response)

**Summary and General Comments**

Reviewer #1: (No Response)

PLOS authors have the option to publish the peer review history of their article (what does this mean?). If published, this will include your full peer review and any attached files.

Reviewer #1: No

---

## [Editor Report · Acceptance letter]

20 Nov 2024

Dear Mr Bugssa,

We are delighted to inform you that your manuscript, "Prevalence and intensity of Schistosoma mansoni infection, and contributing factors in Alamata district of Tigray Region, Northern Ethiopia.," has been formally accepted for publication in PLOS Neglected Tropical Diseases.

Best regards,

Shaden Kamhawi

co-Editor-in-Chief

Paul Brindley

co-Editor-in-Chief
